# Under-Fuelling for the Work Required? Assessment of Dietary Practices and Physical Loading of Adolescent Female Soccer Players during an Intensive International Training and Game Schedule

**DOI:** 10.3390/nu15214508

**Published:** 2023-10-24

**Authors:** Samuel J. McHaffie, Carl Langan-Evans, Juliette A. Strauss, José L. Areta, Christopher Rosimus, Martin Evans, Ruth Waghorn, James P. Morton

**Affiliations:** 1Research Institute for Sport and Exercise Sciences (RISES), Liverpool John Moores University, Byrom Street, Liverpool L3 3AF, UK; s.j.mchaffie@2021.ljmu.ac.uk (S.J.M.);; 2The Football Association, St George’s Park, Newborough Road, Needwood, Burton-Upon-Trent DE13 9PD, UK

**Keywords:** carbohydrate, energy availability, energy intake

## Abstract

Previous studies demonstrate that “under-fuelling” (i.e., reduced carbohydrate (CHO) and energy intake (EI) in relation to recommended guidelines) is prevalent within adult female soccer players, the consequence of which may have acute performance and chronic health implications. However, the dietary practices of adolescent female soccer players, a population who may be particularly at risk for the negative aspects of low energy availability (LEA), are not well documented. Accordingly, we aimed to quantify EI and CHO intake, physical loading and estimated energy availability (EA) in elite national team adolescent female soccer players (*n* = twenty-three; age, 17.9 ± 0.5 years) during a 10-day training and game schedule comprising two match days on day six (MDa) and nine (MDb). The players self-reported their EI via the remote food photography method, whilst the physical loading and associated exercise energy expenditure were assessed via GPS technology. The relative CHO intake was significantly greater (all *p* < 0.05) on the day before the first match (MD-1a) (4.1 ± 0.8 g·kg^−1^), on the day before the second match (MD-1b) (4.3 ± 1.1 g·kg^−1^), MDa (4.8 ± 1.2 g·kg^−1^) and MDb (4.8 ± 1.4 g·kg^−1^) in comparison to most other days (<4 g·kg^−1^). The mean daily measured EA over the 10-day period was 34 ± 12 kcal·kg FFM^−1^·day^−1^ (with six players, i.e., 34%, presenting LEA), though, when adjusting the energy intake for potential under-reporting, these values changed substantially (44 ± 14 kcal·kg FFM^−1^·day^−1^, only one player was classed as presenting LEA). Such data suggest that the prevalence of LEA amongst female team sport athletes may be over-estimated. Nonetheless, players are still likely under-fuelling for the work required in relation to the daily CHO recommendations (i.e., >6 g·kg^−1^) for intensive training and game schedules. These data provide further evidence for the requirement to create and deliver targeted player and stakeholder education and behaviour change interventions (especially for younger athletes) that aim to promote increased daily CHO intake in female soccer players.

## 1. Introduction

In the last decade, there has been a substantial growth in female soccer participation [1], the result of which has likely translated to increased professionalism at the elite level of the sport. Such growth is underpinned by the increased strategic investment from local, continental and national governing bodies [2]. The professionalism of the women’s game has also contributed to the inevitable rise in the provision of staffing at professional clubs, including medical and sport science support [3]. Importantly, however, the evidence base to support the delivery of sport science and medicine services is not nearly comparable to the men’s game and, hence, there have been multiple calls within the academic community for a strategic and targeted approach to research that seeks to improve the health and performance of female players [4]. This is especially the case for the research base that supports the provision of an evidence-based sports nutrition programme, where the energetic requirements of elite female soccer players are now only beginning to be understood.

Indeed, we [5] and others [6] have recently reported the direct assessment of adult players’ energy expenditure (EE), assessed using the gold-standard, doubly labelled water technique. Such reports document absolute energy expenditures of 2693 ± 432 and 2918 ± 322 kcal·day^−1^ in a cohort of English (international competition) and Norwegian (domestic competition) players, respectively. Notwithstanding the potential for dietary under-reporting, it is noteworthy that both studies also highlighted that the players’ self-reported energy intake (EI) was below the recommended values, such that low energy availability (LEA, as classified according to the published values of <30 kcal·kg FFM^−1^·day^−1^) was estimated to occur in 23 [5] and 88% [6] of the players on training days. Such data also agree with the prevalence of LEA that was reported in English players playing in their respective domestic league [7]. Such high prevalence of LEA is especially concerning owing to the potential negative consequences of chronic LEA, as documented in both the female athlete triad [8] and relative energy deficiency in sport (RED-S) [9] models. It is acknowledged, however, that the thresholds for distinguishing LEA are based on short-term laboratory interventions of consistent daily exercise energy expenditure and energy intake, which are not always applicable to ‘real-world’ athletes, given that an athlete’s daily energy availability is not likely to remain constant [10,11]. In this way, there is a possibility that the prevalence of LEA amongst female team sport athletes may have been over-estimated.

In considering the potential reasons underpinning the previously reported dietary practices (especially in relation to sub-optimal daily carbohydrate intake), we recently completed a qualitative investigation exploring the nutrition culture within the women’s game [12]. Importantly, we reported a culture of ‘carbohydrate fear’ whereby players (as also ‘told’ by stakeholders) reported consciously under-consuming (or avoiding) dietary carbohydrate (CHO) intake due to perceived pressures surrounding body composition and body image, as mediated by coach pressure and social media influences. When considered this way, the need for the education of both players and stakeholders becomes readily apparent, the timing of which should likely be targeted to younger players as they transition through the performance pathway from adolescence to adulthood. Indeed, adolescent players may be particularly susceptible to the negative aspects of sub-optimal energy intakes, considering the energy cost of growth and maturation [13]. With this in mind, there is a clear need to extend the study of dietary practices to adolescent players to provide a more complete evaluation of the nutritional practices within the women’s game. The assessment of dietary practices surrounding players’ habitual CHO intake is especially important in the context of considering the role of CHO availability for muscle metabolism and soccer-specific performance. Indeed, a recent assessment of the metabolic demands of female match play using muscle biopsies established that 80% and 70% of type I and type II muscle fibres were classified as empty or almost-empty immediately after a game [14]. Furthermore, there is evidence that low CHO intake may also have a negative impact on technical performance [15].

Accordingly, the aim of the present study was to quantify the energy (and CHO) intake, physical loading and estimated energy availability (EA) of elite adolescent female soccer players. To this end, we studied a cohort of players participating in a 10-day training and game schedule who were representing their national team. Similarly to adult players, we also hypothesised that the players would present dietary practices that are representative of ‘under-fuelling’ for the work required. However, in order to account for the potential of dietary under-reporting, we also reported unadjusted and adjusted estimations of energy availability based on the correction factor recently reported by Dasa et al. [6].

## 2. Materials and Methods

### 2.1. Participants

Twenty-three (*n* = two goalkeepers and *n* = twenty-one outfielders) female soccer players (mean ± SD: age: 17.9 ± 0.5 y, body mass: 61.6 ± 6.1 kg, stature: 168 ± 5 cm) representing the same national team in the under (U) 18 age group, volunteered to take part in the study. The participants’ characteristics categorised by playing position are represented in Table 1. Written informed parental/guardian consent and player assent were obtained for the participants ≤ 17 years old, and the participants ≥ 18 years old provided their own consent. Ethical approval was granted by the Liverpool John Moores University (22/SPS/027).

### 2.2. Study Design

All the players took part in a 10-day international training camp, comprising five training days, three rest days and two match days. The match days occurred on days six and nine; therefore, the last five days could be considered as a “congested fixture” period. All the days are defined relative to the first (a) or second (b) match day (i.e., MD-5, MD-4, MD-3, MD-2, MD-1a, MDa, MD+1a, MD-1b, MDb and MD+1b). Both matches were at “home” against other nations, and an overview of the training and game schedule is shown in Table 2. During the 10-day period, all the players self-reported their energy and macronutrient intake and physical activity, whilst the pitch-based training and match load was measured using a global positioning system (GPS) technology. A bioelectrical impedance analysis (BIA) was also used to assess daily body mass, body composition and body water. All the players were available for selection for both matches, with no injuries preventing any participation within the study.

### 2.3. Pre-Data Collection Education

Six days prior to commencing the data collection period, an initial online education session was held for all the staff members who would be in attendance, in order to inform them of the rationale and study protocol. Three days later, an additional education session was held online for all the players and a representative parent/guardian, where the primary focus was to introduce and educate the players on the remote food photography technique (RFPM). The players arrived “on-site” one day prior to the start of data collection and also took part in a further education session where an ‘in-person’ demonstration of the RFPM method was completed by the lead researcher.

### 2.4. Bioelectrical Impedance Analysis

The body mass and composition were assessed via the BIA (MC-980MA PLUS; Tanita Corp., Tokyo, Japan) in a fasted state each morning. This measurement tool uses a single frequency current of 50 kHz (single frequency BIA [SF-BIA]) and an eight-contact electrode system and was used to assess the body mass (kg), body fat percentage, fat mass (kg), fat-free mass (kg), water mass (kg), water percentage (%) and resting metabolic rate (RMR) of the players. The BIA machine provided a value for the RMR based on the body composition, age and gender of the participants. The players wore the same training kit and removed their shoes and jewellery for the BIA assessments, which were conducted at the same time each day for all the players, between 8 AM and 8:45 AM. All the scans were performed in the fasted state prior to the consumption of food and fluid intake at breakfast.

### 2.5. Quantification of Training and Match Load

The training and match load were measured using a GPS technology (Apex, Statsports, Newry, Northern Ireland), with units worn by all the players, excluding the goalkeepers, for all the pitch-based training sessions and matches. The GPS units were placed inside custom-made manufacturer-provided vests (Apex, STATSports, Newry, Northern Ireland) which were held on the players’ upper-back, between both scapulae, allowing the exposure of the GPS antennae in order to acquire a clear satellite connection. The variables measured include the duration (mins), distance (m), maximum velocity (MV) (m/s), accelerations (>3 m·s^−1^), decelerations (>3 m·s^−1^) and time spent in three different speed zones: zone one (3.46–5.28 m·s^−1^), zone two (5.29–6.25 m·s^−1^) and zone three (≥6.26 m·s^−1^). These categories are commonly used within this population, as established by Park et al. [16].

### 2.6. Quantification of Physical Activity Data

Self-reported physical activity was quantified throughout the 10 days using a self- reported activity diary on a Microsoft Form (Microsoft, Washington, DC, USA). Each participant was sent a link to this form at two time-points throughout the day. The participants were instructed to provide a short description of their physical activity (e.g., ‘walking’ or ‘watching TV’) and a rating of their perceived exertion (RPE) for 30 min periods throughout the day, outside of scheduled activities such as mealtimes, team meetings and training. Each entry was then automatically logged on a Microsoft Excel Spreadsheet (Microsoft, Washington, DC, USA), with each activity being converted into a metabolic-equivalent task (MET) to provide an estimation of the EE and, then, assigned one of the following intensity thresholds based upon its EE value: ‘very light’, ‘light’, ‘moderate’, ‘heavy’, and ‘very heavy’ [17]. The purpose of this was to establish whether any additional physical activity (e.g., gym sessions) needed to be accounted for as exercise energy expenditure (EEE), when estimating the EA.

### 2.7. Quantification of Energy and Macronutrient Intake

The self-reported energy and macronutrient intake were quantified throughout the 10-day period, using the RFPM [18]. This method has previously been validated in adolescent team sport athletes [19]. As the lead researcher was present for the entire training camp, reminders were able to be made easily in person throughout the data collection period. As well as in-person reminders, physical prompts were placed in the dining area to remind the players. Importantly, other staff members were asked to remind the players to take their photos during different mealtimes, in order to reduce the monotony of receiving these reminders from the same source.

As per the protocol, the participants were instructed to take two images, at 90 and 45 degrees, of any food or drink they consumed throughout the ten days, including all the meals and snacks. A third image was also taken of any leftovers, if required. These images were sent to the principal investigator via the phone application Threema (Threema GmbH, Pfäffikon, Switzerland), and a female member of staff was present within each of these group chats, as well as the player and the lead researcher. The participants were also instructed to send a brief description of the food items that they had consumed. The lead researcher constructed two portions (small and large) of each of the foods available, prior to the arrival of the players and staff members for mealtimes. These were weighed and photographed, providing images to compare to during the analysis process, ensuring a more accurate analysis of portion size. The descriptions of foods and drinks consumed before, during and after the training and matches were also sent, as the players did not have access to their phones at this time. Some of the players brought their own snacks to the training camp; however, all the meals and multiple snacks were provided to the players throughout each day, and an outline of the timings can be viewed in Table 2, within the schedule. During the training and matches, the players were also provided with sports drinks if desired, in addition to water, and, post training and match, snacks were also provided (e.g., protein bars, fruit, cereal bars, whey protein, etc.). Once throughout the training camp, every player completed a dietary recall, to check for any missed data. This also provided an opportunity for the lead researcher to feedback on the quality of their data provision and provide any additional education if needed.

The energy and macronutrient intake were analysed by a Sport and Exercise Nutrition register (SENr)-accredited practitioner using the dietary analysis software Nutritics (Nutritics, v5, Dublin, Ireland). With the energy, CHO and protein intake quantified as kilocalories and grams, respectively, in both absolute and relative (to each player’s body mass) terms. During the analysis period, comparisons were made between the photos taken by the players and those of the weighed portion sizes, in order to increase the accuracy of the estimations. To ensure the reliability of energy and macronutrient intake data, a second SENr nutritionist also analysed a sample of food diaries chosen at random (*n* = five, equating to 50 days of entries in total), with the inter-rater reliability determined via an independent *t*-test. No significant differences were observed between the estimations for energy (*p* = 0.96, 95% CI −156 to 38), CHO (*p* = 0.11, 95% CI −31 to 3), protein (*p* = 0.14, 95% CI −12 to 1) and fat (*p* = 0.13, 95% CI −8 to 3). Furthermore, an adjusted EI was calculated by increasing the self-reported EI by 22%, in an attempt to account for the under-reporting that has previously been documented in a comparable sample population [6]. The 22% estimation of under-reporting in Dasa et al. [6] was calculated by comparing the self-reported intake to the calculated energy intake, as inferred from an assessment of energy expenditure (using the DLW technique) and negligible changes in body mass.

### 2.8. During Exercise Energy Expenditure

For field-based sessions, the GPS devices with individualised player descriptives inputted provided a value for the players’ exercise energy expenditure (EEE), which was subsequently increased by 10.7%, based on recent data which demonstrated that this GPS system underestimates EEE within this population [20]. For gym-based sessions, the EEE was estimated based on the activity diaries, with values corrected for everyone using the resting metabolic rate (RMR) data from the BIA machine, whereby the energy expended for RMR during the session was subtracted from the estimation of the EEE.

### 2.9. Estimation of Low Energy Availability and Adjusted Low Energy Availability

The EI and EEE throughout the 10-day period were used to calculate the EA (EA = (EI − EEE)/FFM)) [21] for all the outfield players. To allow for the comparison with the previous literature on female soccer players [5,6,7], the EA was categorised as optimal, (>45 kcal·kg FFM^−1^·day^−1^), reduced (30–45 kcal·kg FFM^−1^·day^−1^) and low (<30 kcal·kg FFM^−1^·day^−1^). An adjusted EA was calculated using the same equation, with the adjusted EI inputted.

### 2.10. Statistical Analysis

All the data were initially assessed for normality of distribution using the Shapiro–Wilk’s test and for outliers using box plots. To determine differences between the days in terms of the absolute and relative energy and macronutrient intake, BIA and GPS data, a one-way between-groups analysis of variance (ANOVA) was used. Where significant main effects were present, an LSD post hoc analysis was conducted to locate specific differences (level of significance set at *p* < 0.05). Ninety-five percent confidence intervals for the difference were also presented. All the statistical analyses were completed using SPSS (version 26; SPSS, Chicago, IL, USA), where *p* < 0.05 was indicative of a statistical significance. All the data are presented as mean ± SD.

## 3. Results

### 3.1. External Load

External loading variables are presented for all the outfield players (*n* = 21) across all the training sessions and games. Figure 1 displays the training and match load metrics for all the days, including the duration, total distance, distance covered within specific speed zones, maximum speed, accelerations and decelerations. No “pitch-based” training sessions occurred on MD-3, MD+1a or MD+1b.

Significant differences were observed between the days for all the variables (all *p* < 0.05), and pair-wise comparisons are denoted in the Figures’ legends. When considered together, the data are suggestive of the traditional approach to micro-cycle periodisation within soccer, such that volume (i.e., duration and total distance) and intensity (i.e., high-intensity distance, maximal speed, accelerations and decelerations) are greater on MD-4, and a gradual daily reduction is evident as the proximity to the first game increases (e.g., MDa). The mean daily EEE during the 10-day period was 442 ± 486 kcal·day^−1^.

### 3.2. Energy and Carbohydrate Intake

The players’ self-reported daily energy and CHO intake is presented in Figure 2. The unadjusted absolute (Figure 2A) and relative (Figure 2C) daily energy intake displayed significant differences between the days (both *p* < 0.001; see Figure legend for the pair-wise comparisons), such that the mean values of 2053 ± 486 kcal·day^−1^ (range: 1697–2413 kcal·day^−1^) and 33.6 ± 8.4 kcal·kg·day^−1^ (range: 27.9–39.4 kcal·kg·day^−1^) were reported, respectively.

When accounting for potential dietary under-reporting, a similar pattern of nutritional periodisation was evident for both the adjusted absolute (Figure 2B) and relative energy intake (Figure 2D), with significant differences between the days (both *p* < 0.001; see Figure legend for the pair-wise comparisons). However, the mean absolute and relative energy intake values were now increased to 2505 ± 490 kcal·day^−1^ (range: 2070–2944 kcal·day^−1^) and 40.9 ± 9.1 kcal·kg·day^−1^ (range: 34–48.1 kcal·kg·day^−1^), respectively.

Both the absolute (Figure 2E) and relative (Figure 2F) CHO intake also displayed significant differences between the days (both *p* < 0.001; see Figure legend for the pair-wise comparisons). The most prominent difference was reported as a greater relative CHO intake on the days before the game (4.8 ± 1.1 g·kg^−1^ and 4.3 ± 1.1 g·kg^−1^, reported on MD-1a and MD-1b, respectively) and on game days (4.8 ± 1.2 g·kg^−1^ and 4.8 ± 1.4 g·kg^−1^, reported on MDa and MDb, respectively) when compared to the remaining training and rest days. No formal calculation of an adjusted CHO intake was conducted owing to the difficulty of attributing the potential under-reporting of energy intake to a specific macronutrient.

### 3.3. Body Composition

The changes in body mass, body composition and body water are presented in Figure 3. Although the fat mass displayed no changes between the days (Figure 3C, *p* = 0.113), all the other variables displayed significant fluctuations throughout the 10-day period (all *p* < 0.05).

In relation to the body mass, significantly greater values were observed on MDa (61.9 ± 6.3 kg) and MDb (61.9 ± 6.2 kg) compared to MD-5 (61.4 ± 6.3 kg; 95% CI = 0.2 to 0.6 kg; *p* < 0.001, 95% CI = 0.1 to 0.6 kg; *p* = 0.002, respectively), MD-4 (61.8 ± 6.5 kg; 95% CI = 0.04 to 0.5 kg; *p* = 0.021, 95% CI = 0.02 to 0.5 kg; *p* = 0.037, respectively) and MD-2 (61.6 ± 6.3 kg; 95% CI = 0.1 to 0.6 kg; *p* = 0.003, 95% CI = 0.1 to 0.5 kg; *p* = 0.007, respectively). The body mass was also significantly higher on MDa than on MD-3 (61.8 ± 6.4 kg; 95% CI = 0.01 to 0.5 kg; *p* = 0.038), and it was significantly higher on MD+1b (61.8 ± 6.2 kg) than on MD-5 (95% CI = 0.002 to 0.5 kg; *p* = 0.048).

The body fat percentage was significantly higher on MD-5 (25.6 ± 3.3%), MD-4 (25.6± 3.2%), MD-3 (25.7 ± 3.3%), MD-2 (25.7 ± 3.2%), MD-1b (25.8 ± 3.2%) and MDb (25.8 ± 3.4%) than on MD+1b (25.2 ± 0.7%) (all *p* < 0.05).

The fat-free mass was significantly higher on MDa (46.1 ± 4.1 kg) and MD+1b (46.1 ± 4.1 kg), than on MD-5 (45.6 ± 4.2 kg), MD-4 (45.9 ± 4.2 kg), MD-3 (45.8 ± 4.2 kg), MD-2 (45.7 ± 4.1 kg) and MD-1b (45.7 ± 4.0 kg) (all *p* < 0.05).

The water mass and water percentage were both significantly lower on MD-5 (33.0 ± 2.9 kg and 54.0 ± 3.3%, respectively) than on every other day (all *p* < 0.05). The water mass was also significantly higher on MDa (33.7 ± 0.6 kg) than on MD-2 (33.4 ± 3.0 kg; 95% CI = 0.002 to 0.5 kg; *p* = 0.048). It was also significantly higher on MD+1b (34 ± 2.9 kg) than on MD-2 (33.4 ± 3.60 kg; 95% CI = 0.002 to 0.5 kg; *p* = 0.048) and MD-1b (33.4 ± 2.9 kg; 95% CI = 0.002 to 0.5 kg; *p* = 0.048). The water percentage was significantly higher on MD+1b (54.7 ± 2.8 kg) than on MD-2 (54.3 ± 2.5 kg; 95% CI = 0.002 to 0.5 kg; *p* = 0.048), MD-1b (54.3 ± 2.7 kg; 95% CI = 0.002 to 0.5 kg; *p* = 0.048) and MDb (54.3 ± 2.7 kg; 95% CI = 0.002 to 0.5 kg; *p* = 0.048).

### 3.4. Estimated Energy Availability

The estimated daily EA throughout the 10-day period and the mean daily EA for the entire 10-day period are presented in Figure 4. Both the unadjusted and adjusted daily EA (see Figure 4A,C, respectively) showed a significant difference between the days (*p* < 0.05), with the pair-wise comparisons being denoted in the Figure’s legend.

The unadjusted mean daily EA was 34 ± 12 kcal·kg FFM^−1^·day^−1^ within the outfield players, with a prevalence of two players (9%) with an optimal EA, thirteen (57%) players with a reduced EA and six players (34%) with LEA. However, when the energy intake was adjusted for potential under-reporting, the prevalence of LEA changed considerably, such that eight (38%), twelve (57%) and one (5%) players now presented optimal EA, reduced EA, and LEA, respectively (see Figure 4B,D, respectively).

## 4. Discussion

On the basis of recent reports of under-fuelling, LEA and a culture of CHO-fear amongst adult professional female soccer players [5,6,7,12], we aimed to extend our study of the habitual dietary practices of female players to an adolescent population. By studying a cohort of elite players who were training and competing for their national team over a 10-day period, we assessed their self-reported energy intake, physical loading and estimated energy availability. Although the thresholds for distinguishing LEA amongst athletic populations (under real-world, free-living conditions) remain the topic of intense investigation, our data are in agreement with previous reports about adult players [5,6,7] that are suggestive of LEA (*n* = six, i.e., 34% of players presented LEA). However, when accounting for potential dietary under-reporting [6], the pattern of LEA changes substantially, such that only one player was categorised as having LEA (albeit this was using thresholds derived from laboratory-based research models). When considering the issues of under-reporting alongside the problems associated with distinguishing LEA, such data suggest that the ‘true’ prevalence of LEA amongst female team sport athletes may be over-estimated within the literature. Nonetheless, evaluations of daily CHO intake are still suggestive of ‘under-fuelling for the work required’, as based on an assessment of daily CHO intake in relation to the published recommendations for intake during congested fixture schedules (i.e., the last five days of assessment where the two games were played).

To address our aim, we studied the players throughout a national ‘training camp’ environment during which the players trained and resided at the same hotel. Importantly, the training and game schedule comprised the traditional micro-cycle approach that is inherent to soccer, where a game is usually preceded by four or five training days. In this regard, the evaluation of external loading prior to game one (i.e., MDa) demonstrates the typical periodisation of training load that has previously been reported in both adult female [22] and male players [23]. Indeed, we observed that the highest volume (e.g., duration and total distance) and intensity of training (e.g., distance covered within speed zone three, maximal speed and number of accelerations and decelerations) typically occurred four days prior to MDa. In the subsequent days, the training load displayed an apparent tapering of volume and intensity, likely a conscious decision by the coaching team to promote the players’ readiness for the upcoming game. The rest of the data collection period could be considered to more closely resemble a congested fixture schedule whereby two games were played in a four-day period. As such, the two-day period between the games presented as an initial rest day and a second day of reduced training load. Although it is difficult to compare markers of training intensity between studies (owing to the variety of speed thresholds used in the literature), it is noteworthy that the average training and MD total distance reported here (3789 ± 1375 and 6667 ± 3764 m, respectively) are lower than our previous report from adult players (4838 and 6837 m, respectively), who were also participating in a similar international training and game schedule [5]. Unfortunately, we are unable to provide any insight as to whether this is a conscious and planned approach to training that takes into account the adolescent nature of the players and/or whether this population is physically unable to attain the same markers of volume and intensity as the adult players. It is important to note that match day values include the warm-up and all players, including unused substitutes, so this value is heavily impacted by squad sizes across match days.

In relation to the self-reported energy intake, we observed similar absolute values 2053 ± 486 kcal·day^−1^ (range: 1697–2413 kcal·day^−1^) as to those reported previously by our group [5] (1923 ± 357 kcal·day^−1^, range: 1639–2172) and others [6,7] (2274 ± 450 kcal·day^−1^ and 2124 ± 444 kcal·day^−1^, respectively) when assessing adult players competing at both the international and domestic level. On the basis of evaluations of the exercise-related energy expenditure, we subsequently report a mean estimated energy availability of 34 ± 12 kcal·kg FFM^−1^·day^−1^, such that 9, 57 and 34% of the players could be classified as having optimal EA, reduced EA and LEA, respectively. Such prevalence of LEA agrees favourably with that reported by Moss et al., 2020 [7], where 23% of adult players playing in the Women’s Super League of England were categorised as having LEA over a 5-day period (in both cases the exercise energy expenditure was quantified via GPS). However, our data are substantially lower than our previous reports from adult players also competing at the international level, where we reported 88% of the players as having LEA [5]. Such differences between the studies are likely to be due to methodological differences, given that our previous approach for quantifying exercise energy expenditure was based on assessments of ‘activity energy expenditure’ derived from insights from a doubly labelled water assessment of total daily energy expenditure. In this way, the prevalence of LEA was likely to have been over-estimated.

We readily acknowledge the difficulties of accurately quantifying energy availability owing to the technical challenges of accurately assessing both the energy intake and exercise energy expenditure. Indeed, this field is further complicated in that the categorisation of LEA (as <30 kcal·kg FFM^−1^·day^−1^) is based on laboratory studies with homogenous patterns of energy intake and exercise-related expenditure [10]. In contrast, athletes living under free-living conditions typically present daily variations in training volume and intensity (and, hence, exercise energy expenditure), albeit their energy intake may not be adjusted accordingly [11]. Additionally, this field is perhaps most complicated by dietary under-reporting and, as such, symptoms associated with the female athlete triad and RED-S models may have been potentially over-interpreted within the team sport literature. Indeed, a recent study employing a doubly labelled water method to assess the total daily energy expenditure in a cohort of adult Norwegian players [6] reported a discrepancy between the energy expenditure (2918 ± 322 kcal·day^−1^) and the self-reported energy intake (2274 ± 450 kcal·day^−1^) of approximately 22%. In applying the same correction factor to the data presented here (considering that the body mass did not decrease during the study period), we were able to highlight an adjusted mean energy intake of 2505 ± 490 kcal·day^−1^ and an adjusted energy availability of 44 ± 14 kcal·kg FFM^−1^·day^−1^. In this way, the pattern of LEA changed substantially, such that only one player presented LEA according to the adjusted intake. Clearly, further assessments of the total daily energy expenditure in adolescent players (using the DLW method) are now justified for providing further insights on the prevalence of LEA amongst team sport athletes.

It is difficult to provide an adjusted assessment of daily CHO intake owing to the difficulty of attributing dietary under-reporting to a specific macronutrient. However, even when considering the potential under-reporting of CHO intake, our data are still suggestive of elite female soccer players under-consuming CHO in relation to the recommended guidelines. Indeed, the latest UEFA expert group statement on ‘nutrition for football’ recommends at least 6–8 g·kg^−1^ to be consumed daily during periods of congested fixtures and intense training schedules [24]. Additionally, it is well documented that high CHO intakes should be consumed on the day before and after match play, to load and recover muscle glycogen stores, respectively. Indeed, although we observed some evidence supporting some principles of CHO loading (e.g., increased body mass on both match days that may be potentially reflective of increased glycogen storage), the reported daily CHO intake still falls short of the recommended values. Although these guidelines are generic and not specific to female players, recent assessments of the metabolic demands of female match play using muscle biopsies further demonstrate the importance of CHO availability given that 80 and 70% of type I and type II muscle fibres were classified as empty or almost empty immediately after a game [14]. When considered this way, it is likely that the majority of the players studied here commenced both games with sub-optimal muscle glycogen stores, the result of which could compromise both the physical [14] and technical [15] elements of performance. Indeed, of the total daily dietary assessments completed in this study (i.e., 230), there were only 19 (8%) accounts of a player reporting a daily CHO intake >6 g·kg^−1^. Furthermore, although the concept of CHO periodisation is gaining increased acceptance as a targeted nutritional strategy for adult athletes [25], it is unlikely that such periodisation should be recommended here when considering the adolescent nature of the players and the intensive demands of the training and game schedule.

We were unable to ascertain the likely reasons for the apparent under-consumption of CHO intake reported here and whether this was a conscious or unconscious decision by the players. In using a qualitative methodology to explore the nutrition culture within the women’s game (inclusive of both adult and adolescent players), we previously reported that players may under-fuel due to a lack of awareness of nutrition guidelines and, hence, do not readily appreciate the importance of CHO for football development and performance [12]. However, notwithstanding the need for increasing education amongst players and stakeholders, we also reported incidences of intentional under-fuelling due to perceived pressures surrounding body composition assessments and body image issues. In considering such insights through the lens of behavioural change models such as the COM-B framework [26,27], it is apparent that the social opportunity (i.e., cultural norms) is not yet conducive to permit the repeated nutritional behaviours that could be considered representative of a positive nutrition and fuelling culture. As such, there is a need for further multi- and inter-disciplinary research that addresses the barriers and enablers to improve the provision of nutrition services within the women’s game. The need for targeted education is especially relevant here, given the adolescent nature of the players and the potential to instil sound nutritional habits and behaviours as they transition into adulthood and the professional game.

As with all dietary assessment studies, the present study is not without limitations. Indeed, the present paper is based on one team only and our data cannot be generalised to other female soccer teams. Furthermore, and as alluded to previously, inferences around energy availability and sub-optimal CHO intakes are largely based on laboratory studies where, in the context of the latter, current recommendations are largely based on male athletes. The present study also did not evaluate the effects of the players’ habitual dietary practices on associated performance metrics, fatigue indices or any symptomology associated with the RED-S or female athlete triad models. When considered this way, it is clear that a strategic approach to further research is now required to more accurately inform the creation of evidence-based nutritional guidelines for this population.

## 5. Conclusions

In summary, the present data provide the first report assessing the habitual dietary practices of a cohort of adolescent players currently playing at the highest level of the game (i.e., the international standard). Although we acknowledge the potential for dietary under-reporting, our data are still suggestive of the players under-fuelling for the work required. This is especially the case in relation to the total daily CHO intake during times of intense training and competition.

## Figures and Tables

**Figure 1 nutrients-15-04508-f001:**
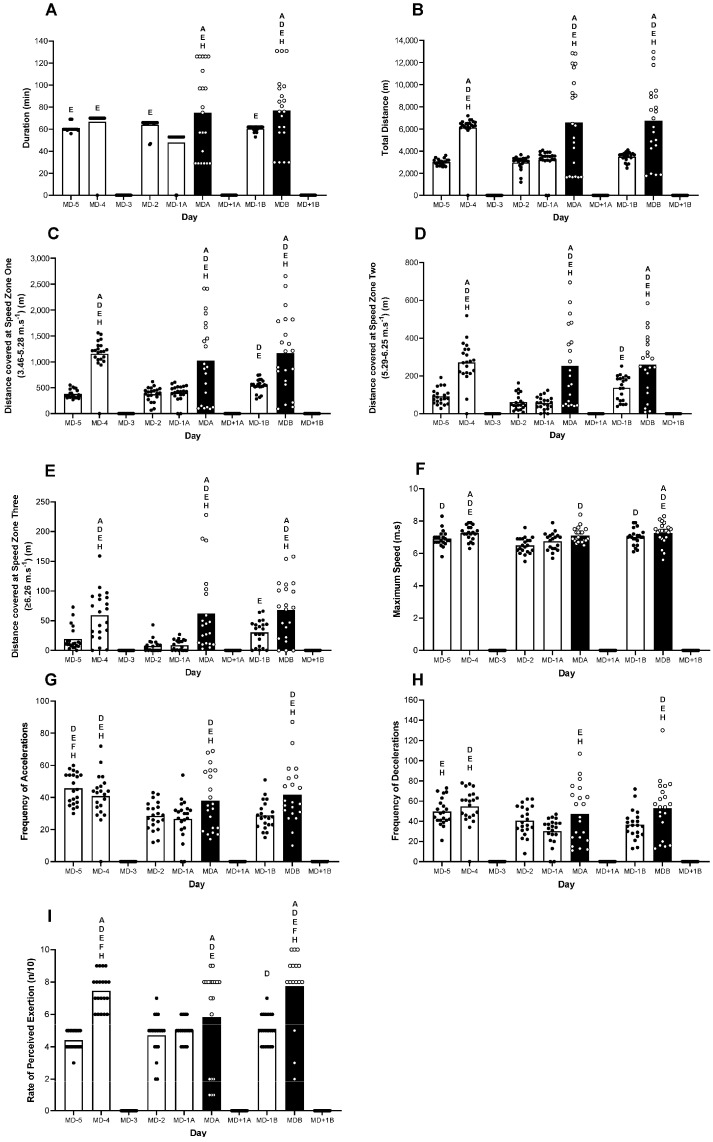
An overview of external loading throughout the 10-day period. Pitch-based training and match’s (**A**) duration, (**B**) total distance, (**C**) distance at speed zone one (3.46–5.28 m·s^−1^), (**D**) distance at speed zone two (5.29–6.25 m·s^−1^), (**E**) distance at speed zone three (≥6.26 m·s^−1^), (**F**) maximum speed, (**G**) frequency of accelerations, (**H**) frequency of decelerations and (**I**) rate of perceived exertion. Black bars represent match days. A is significantly higher than MD5, B is significantly higher than MD-4, C is significantly higher than MD-3, D is significantly higher than MD-2, E is significantly higher than MD-1a, F is significantly higher than MDa, G is significantly higher than MD+1a, H is significantly higher than MD-1b, I is significantly higher than MDb and J is significantly higher than MD+1b, all *p* < 0.05.

**Figure 2 nutrients-15-04508-f002:**
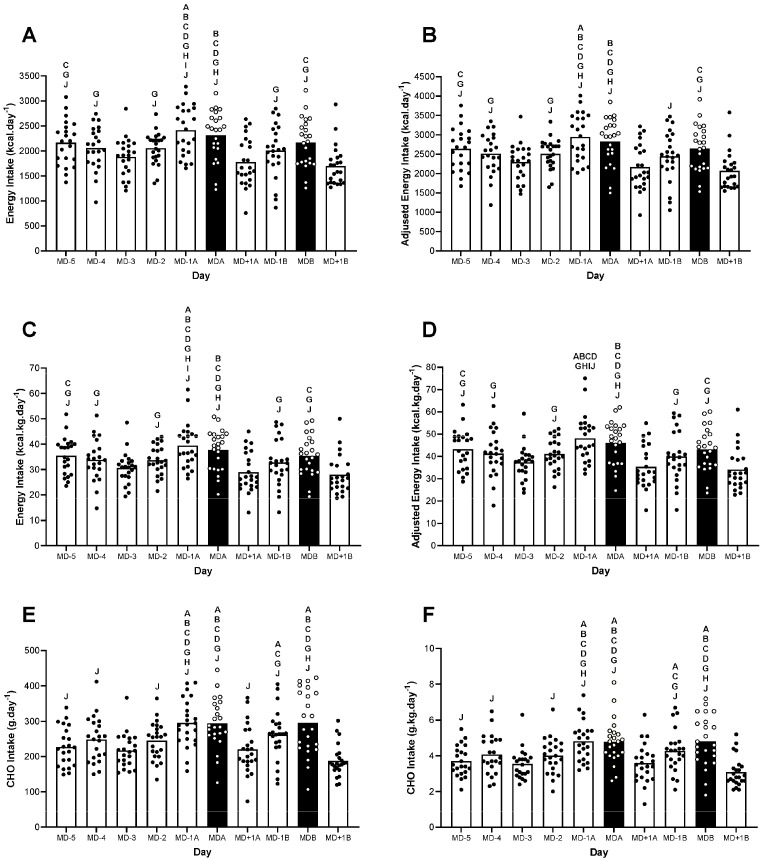
An overview of the energy and CHO intake throughout the 10-day period. (**A**) Absolute energy intake, (**B**) adjusted absolute energy intake, (**C**) relative energy intake, (**D**) adjusted relative energy intake, (**E**) absolute carbohydrate intake and (**F**) relative carbohydrate intake. Black bars represent match days. A is significantly higher than MD-5, B is significantly higher than MD-4, C is significantly higher than MD-3, D is significantly higher than MD-2, E is significantly higher than MD-1, F is significantly higher than MDa, G is significantly higher than MD+1a, H is significantly higher than MD-1b, I is significantly higher than MDb and J is significantly higher than MD+1b, all *p* < 0.05.

**Figure 3 nutrients-15-04508-f003:**
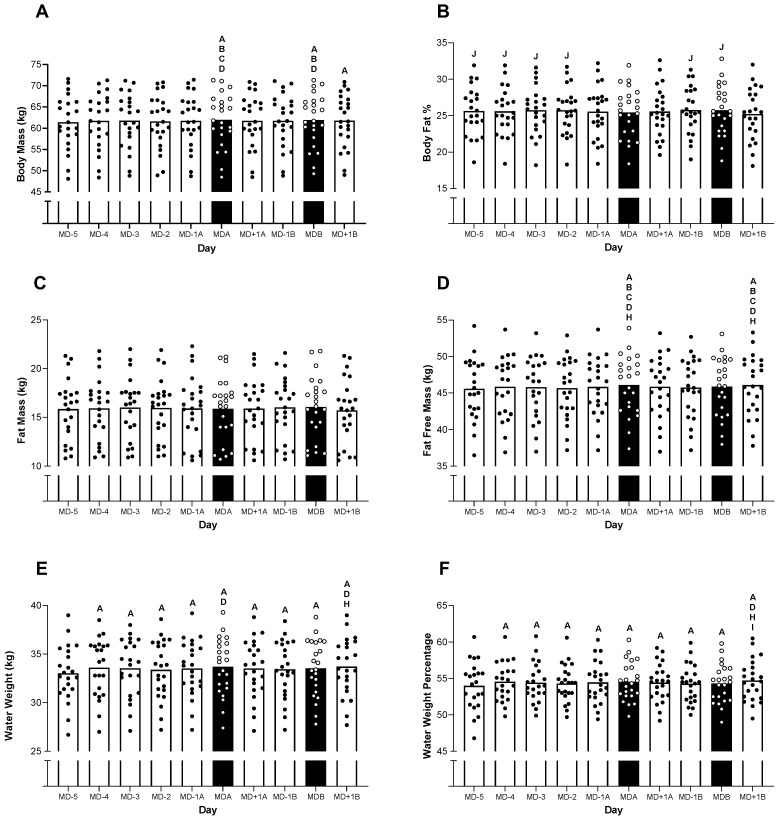
An overview of the BIA data throughout the 10-day period. (**A**) Body mass, (**B**), body fat percentage, (**C**) fat mass, (**D**) fat-free mass, (**E**) water mass and (**F**) water mass percentage. Black bars represent match days. A is significantly higher than MD-5, B is significantly higher than MD-4, C is significantly higher than MD-3, D is significantly higher than MD-2, E is significantly higher than MD-1, F is significantly higher than MDa, G is significantly higher than MD+1a, H is significantly higher than MD-1b, I is significantly higher than MDb and J is significantly higher than MD+1b, all *p* < 0.05.

**Figure 4 nutrients-15-04508-f004:**
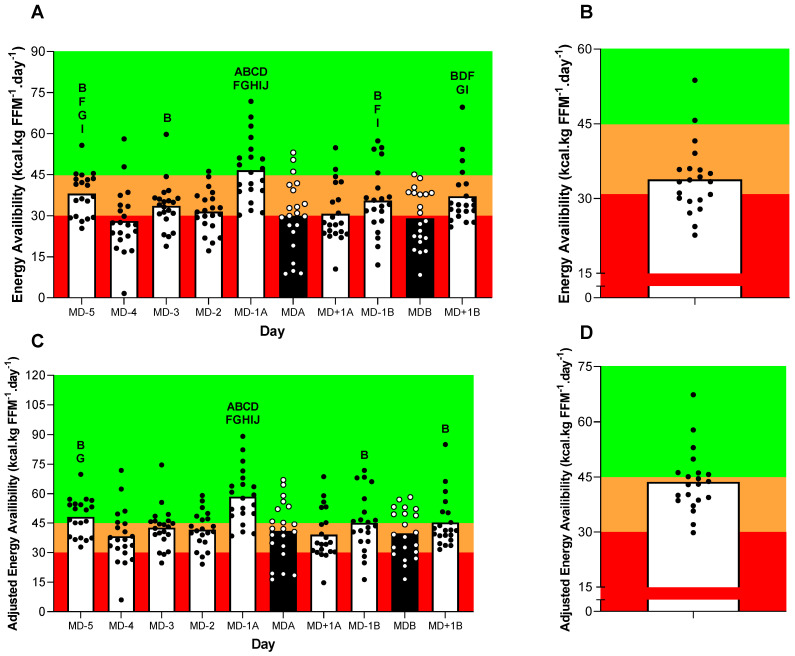
An overview of the energy availability and adjusted energy availability throughout the 10-day period. (**A**) Energy availability across 10 days, (**B**) average daily energy availability, (**C**) adjusted energy availability across 10 days and (**D**) average daily adjusted energy availability. Black bars represent match days. A is significantly higher than MD-5, B is significantly higher than MD-4, C is significantly higher than MD-3, D is significantly higher than MD-2, E is significantly higher than MD-1, F is significantly higher than MDa, G is significantly higher than MD+1a, H is significantly higher than MD-1b, I is significantly higher than MDb and J is significantly higher than MD+1b, all *p* < 0.05. The colours represent the optimal, (>45 kcal·kg FFM^−1^·day^−1^), reduced (30–45 kcal·kg FFM^−1^·day^−1^) and low (<30 kcal·kg FFM^−1^·day^−1^) categories of EA.

**Table 1 nutrients-15-04508-t001:** Baseline participants’ characteristics.

	Goalkeepers (*n* = Two)	Defenders (*n* = Eight)	Midfielders (*n* = Five)	Attackers (*n* = Eight)	Whole Squad (*n* = Twenty-Three)
Age (years)	16.5 ± 0.7	16.4 ± 0.5	16.4 ± 0.5	16.4 ± 0.7	17.9 ± 0.5
Stature (cm)	169 ± 2.8	172 ± 4.2	165 ± 6.4	166 ± 4.5	168 ± 5
Body Mass (kg)	62.8 ± 4.8	60.4 ± 4.6	59.1 ± 8.2	64 ± 6.9	61.6 ± 6.1
Fat-Free Mass (kg)	46.0 ± 4.1	46.0 ± 3.4	43.5 ± 5.3	46.6 ± 4.5	45.7 ± 4.2
Fat Mass (kg)	16.8 ± 0.7	14.4 ± 2.5	15.6 ± 3.5	17.4 ± 3.5	15.9 ± 3.1
Body Fat (%)	26.8 ± 0.9	23.8 ± 3.0	26.2 ± 3.2	27 ± 3.4	25.7 ± 3.3

**Table 2 nutrients-15-04508-t002:** An overview of the training and game schedule, with training sessions, matches and meal times highlighted in bold.

	Match Day −5	Match Day −4	Match Day −3	Match Day −2	Match Day −1	Match Day	Match Day +1	Match Day −1	Match Day	Match Day +1
**09.00**	**Breakfast**	**Breakfast**	**Breakfast**	**Breakfast**	**Breakfast**	**Breakfast**	**Breakfast**	**Breakfast**	**Breakfast**	**Breakfast**
**10.00**	Rest	Rest	**Gym**	Rest	Rest	Rest	Rest/Recovery	Rest	**Breakfast**	Rest
**11.00**	**Training (pitch)**	**Training (pitch)**	Rest/Education	**Training (pitch)**	**Training (pitch)**	**Pre-match meal**	Rest/Recovery	**Training (pitch)**	**Prep and warm-up**	Rest
**12.00**	**Training (pitch)**	**Training (pitch)**	Rest/Education	**Training (pitch)**	**Training (pitch)**	Rest	Rest/Recovery	**Training (pitch)**	**Match**	**Lunch**
**13.00**	Snack	Snack	**Lunch**	**Snack**	**Snack**	**Prep and warm-up**	**Lunch**	**Snack**	**Match**	Players depart
**14.00**	**Lunch**	**Lunch**	Rest/Meeting	**Lunch**	**Lunch**	**Match**	Rest/Recovery	**Lunch**	**Snack**	Players depart
**15.00**	Rest/Education	Rest	Rest/Meeting	Rest/Education	Rest/Education	**Match**	**Gym**	Rest/Education	**Post-match meal**	Players depart
**16.00**	Rest	Rest	Rest/Meeting	Rest	Rest	**Snack**	**Snack**	Rest/Education	Rest	Players depart
**17.00**	Rest	Rest	Rest/Meeting	**Gym**	Rest	**Post-match meal**	Rest/Education	Rest	Rest	Players depart
**18.00**	Rest	Rest	Rest/Meeting	Rest	Rest	Rest	Rest/Education	Rest	Rest	Players depart
**19.00**	**Dinner**	**Dinner**	**Dinner**	**Dinner**	**Dinner**	Rest	**Dinner**	**Dinner**	**Dinner**	Players depart
**20.00**	Rest	Rest/Education	Rest/Education	Rest/Education	Rest	Rest	Rest	Rest	Rest	Players depart
**21.00**	**Snack**	**Snack**	**Snack**	**Snack**	**Snack**	**Snack**	**Snack**	**Snack**	**Snack**	Players depart
**22.00**	Rest	Rest	Rest	Rest	Rest	Rest	Rest	Rest	Rest	Players depart

## Data Availability

Data supporting the reported results are available on request.

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
