# Peer review of "Under-Fuelling for the Work Required? Assessment of Dietary Practices and Physical Loading of Adolescent Female Soccer Players during an Intensive International Training and Game Schedule"

_nutrients, 2023, doi:10.3390/nu15214508_

Round 1

Reviewer 1 Report

The study demonstrates a significant level of academic rigor and a deep dedication to your field of study.  I believe that the research has the potential to make a valuable contribution to the scientific community.  I examined the article and aimed to provide constructive feedback to help further enhance it.

Abstract:

Please clarify the acronyms in the abstract: EA; MD-1a; and MD-1b.

Introduction:

Explain the meaning of the acronym EI (line 55).

I suggest referencing the sources again in the passage: "...was estimated to occur in 23% and 88% of players on training days, respectively (?). (lines 56 and 57).

Material and Methods:

Please include the approval report number (or other evidence) from the ethics committee (line 99).

Provide additional information about the BIA protocol. What were the pre-assessment procedures? When was the assessment conducted during the day? Was there control over food and liquid intake?

Explain the acronym EEE (line 157).

Justify the 22% adjustment in EI. In the study by Dasa et al. (2023) [6], this value is approximate and reflects the reality of the assessed population (professional female soccer players). Therefore, it may not be applicable to the population under scrutiny in this study (adolescent female soccer players). Additionally, in the Dasa et al. study, the difference between energy expenditure and reported energy intake was determined, rather than a comparison between actual intake and reported intake. Is this interpretation correct?

Also, provide a rationale for the 10.7% adjustment in exercise energy expenditure. Why was there no comparison between the adjusted and unadjusted values, as was done for EI?

Results:

I recommend presenting the exercise energy expenditure mean value. These data are crucial for comprehending the results of EA and may be important for future studies on similar topics. If this is carried out, it is imperative to discuss the results, even if briefly.

Discussion:

Make the study's limitations more explicit. I suggest dedicating a paragraph to outlining these limitations.

It is crucial to clarify that the classifications of EA and of the CHO intake results were established based on literature values. Parameters related to performance or physical health were not evaluated. It should be suggested that studies be conducted to advance in this direction, aiming to verify whether the presented parameters are indeed appropriate or not.

Conclusion:

The conclusion should be rewritten to exclusively focus on the studies objectives.

Summarizing the study's execution and its significance should be reserved for the discussion section.

Reviewer 2 Report

Attached.
